# A Potential Biomarker of Dynamic Change in Peripheral CD45RA^−^CD27^+^CD127^+^ Central Memory T Cells for Anti-PD-1 Therapy in Patients with Esophageal Squamous Cell Carcinoma

**DOI:** 10.3390/cancers15143641

**Published:** 2023-07-16

**Authors:** Mei Sakuma, Kosaku Mimura, Shotaro Nakajima, Akinao Kaneta, Tomohiro Kikuchi, Azuma Nirei, Takeshi Tada, Hiroyuki Hanayama, Hirokazu Okayama, Wataru Sakamoto, Motonobu Saito, Tomoyuki Momma, Zenichiro Saze, Koji Kono

**Affiliations:** 1Department of Gastrointestinal Tract Surgery, Fukushima Medical University School of Medicine, Fukushima 960-1295, Japan; 2Department of Blood Transfusion and Transplantation Immunology, Fukushima Medical University School of Medicine, Fukushima 960-1295, Japan

**Keywords:** esophageal squamous cell carcinoma, biomarker, central memory T cell, PD-1, TIM-3

## Abstract

**Simple Summary:**

In order to develop a biomarker predicting the efficacy of chemotherapy (CT), chemoradiotherapy (CRT), and nivolumab therapy (NT) for patients with esophageal squamous cell carcinoma (ESCC), we evaluated the subpopulation of T cells in ESCC patients treated with each therapy. The frequencies of PD-1^+^ or TIM-3^+^CD4^+^ T cells were significantly higher in patients with cStage IV. PD-1^+^CD4^+^ and TIM-3^+^CD8^+^ T-cell populations were significantly higher in patients treated with CRT but were not associated with treatment response. The frequencies of both CD4^+^ and CD8^+^ central memory T cells (T_CM_) were significantly decreased during NT in the progressive disease group. Taken together, the alteration in frequency of T_CM_ during NT may be a biomarker to predict its therapeutic response in ESCC patients.

**Abstract:**

In order to develop a biomarker predicting the efficacy of treatments for patients with esophageal squamous cell carcinoma (ESCC), we evaluated the subpopulation of T cells in ESCC patients treated with chemotherapy (CT), chemoradiotherapy (CRT), and nivolumab therapy (NT). Fifty-five ESCC patients were enrolled in this study, and peripheral blood samples were collected before and after CT or CRT and during NT. Frequencies of memory, differentiated, and exhausted T cells were evaluated using flow cytometry among cStages, treatment strategies, pathological responses of CT/CRT, and during NT. The frequencies of PD-1^+^ or TIM-3^+^CD4^+^ T cells were significantly higher in patients with cStage IV. PD-1^+^CD4^+^ and TIM-3^+^CD8^+^ T-cell populations were significantly higher in patients treated with CRT but were not associated with treatment response. The frequencies of both CD4^+^ and CD8^+^ CD45RA^−^CD27^+^CD127^+^ central memory T cells (T_CM_) were significantly decreased during the course of NT in the progressive disease group. Taken together, the alteration in frequency of CD45RA^−^CD27^+^CD127^+^ T_CM_ during NT may be a biomarker to predict its therapeutic response in ESCC patients.

## 1. Introduction

Esophageal cancer is the sixth leading cause of cancer-related deaths and seventh most frequent cancer worldwide (GLOBOCAN 2020) [1]. Although advanced esophageal cancer patients are treated with multidisciplinary modalities, including surgery, chemotherapy (CT), and radiotherapy, the 5-year survival rate for patients who can undergo esophagectomy remains 59.3% in Japan [2,3]. For example, the response rates of cisplatin and 5-fluorouracil (CF) chemotherapy, and of docetaxel, cisplatin, and 5-fluorouracil (DCF) chemotherapy in esophageal squamous cell carcinoma (ESCC) have been reported to be 33.3% and 62%, respectively [4,5]. Recently, the Checkmate-649 trial revealed that nivolumab plus CF-chemotherapy or nivolumab plus ipilimumab have been shown to have better survival outcomes than CF-chemotherapy alone in advanced ESCC patients [6], and it is currently recommended that nivolumab plus CF-chemotherapy or nivolumab plus ipilimumab is the first line treatment for advanced ESCC patients. However, in order to further improve the efficacy of multidisciplinary treatments, it is important to develop biomarkers for predicting their efficacy and analyze the mechanisms of their therapeutic effects.

It has been reported that CT and chemoradiotherapy (CRT) under certain conditions can induce immunogenic tumor cell death in the tumor microenvironment, resulting in activation of anti-tumor immune responses [7,8,9,10], and we previously reported that CRT induced tumor-antigen-specific T cells in patients with advanced ESCC [11]. Therefore, collaboration between CT/CRT and immunotherapy based on T-cell immunity could potentially induce synergistic effects, leading to improved clinical efficacy.

It is well known that exhausted T cells express immune inhibitory receptors, including the programmed cell death protein-1 (PD-1) and T-cell immunoglobulin-3 (TIM-3) [12]. Immunotherapy with monoclonal antibodies targeting these immune checkpoint receptors or ligands can rescue the immune function of T cells [13]. T cells were also classified into naïve, effector, and memory T-cell subpopulations [14], and memory T cells are thought to play an important role in anti-tumor immunity compared with naïve and effector T cells [15,16]. In order to develop a biomarker that can predict the efficacy of CT, CRT, and immunotherapy, it is reasonable to analyze the frequencies of exhausted and memory T cells affected by each treatment. Although there are many markers for memory T cells [14,15,16,17,18], Fairfax et al. classified memory T cells using CD45RA and CD27 in central memory T cells (T_CM_, CD45RA^−^CD27^+^ T cells) and effector memory T cells (T_EM_, CD45RA^−^CD27^−^ T cells) [18], and Martin et al. reported that CD127 was expressed on both T_CM_ and T_EM_ [16]. Therefore, we focused on CD45RA, CD27, and CD127 markers to detect T_CM_ and T_EM_ in the present study.

To develop a biomarker predicting the efficacy of CT, CRT, and nivolumab treatment, the present study aimed to evaluate the frequencies of memory and exhausted T cells in ESCC patients after treatment with CT or CRT, in addition to during nivolumab treatment.

## 2. Materials and Methods

### 2.1. Patients and Peripheral Blood Samples Collection

ESCC patients who had been treated at the Gastrointestinal Tract Surgery department of Fukushima Medical University Hospital between January 2021 and May 2022 were included in the present study. Clinical stage (cStage) was diagnosed according to the Japanese Classification of Esophageal Cancer (11th Edition). In principle, according to the esophageal cancer practice guidelines, 2017, edited by the Japan Esophageal Society, patients with cStage II or III were treated with neoadjuvant therapy, CF-chemotherapy with/without radiotherapy or DCF-chemotherapy, and followed by curative esophagectomy. Patients with cStage I were treated with esophagectomy without neoadjuvant therapy, and those with cStage IV were treated with CT or CRT as the first-line treatment, followed by nivolumab treatment as the second line.

In cStage I patients, peripheral blood samples were collected prior to surgery without neoadjuvant therapy. In cStage II or III patients, neoadjuvant treatment with CT or CRT were completed and then blood samples were collected prior to surgery. In patients with cStage IV, peripheral blood samples were collected prior to nivolumab treatment and just before nivolumab administration at the time when CT scan was performed to determine the initial treatment efficacy of nivolumab. Peripheral blood mononuclear cell (PBMC) samples were stored in a liquid nitrogen tank, and plasma samples were stored in a −80-degree freeze.

### 2.2. Neutrophil-to-Lymphocyte Ratio

Neutrophil-to-lymphocyte ratio (NLR) was defined as absolute neutrophil count divided by the absolute lymphocyte count. We used neutrophil and lymphocyte counts from medical records when peripheral blood samples were collected.

### 2.3. Flow Cytometry

The collected PBMCs were used for flow cytometry and the manufacturer’s recommended concentration of each antibody was used for staining. For analysis of memory T-cell markers, the cells were stained for 30 min at 4 °C in the dark with antibodies, including APC/Cyanine7 mouse anti-human CD3 mAb (BioLegend, San Diego, CA, USA), FITC mouse anti-human CD4 mAb (BioLegend), Pacific Blue mouse anti-human CD8a mAb (BioLegend), PE/Cyanine7 mouse anti-human CD45RA mAb (BioLegend), APC mouse anti-human CD27 mAb (BioLegend), and PE mouse anti-human CD127 mAb (BioLegend) in the same tube. For analysis of exhaustion and differentiation markers on T cells, the cells were stained for 30 min at 4 °C in the dark with antibodies including APC/Cyanine7 mouse anti-human CD3 mAb, FITC mouse anti-human CD4 mAb, Pacific Blue mouse anti-human CD8a mAb, APC mouse anti-human CD279 (PD-1) mAb (BioLegend), PE mouse anti-human CD366 (TIM-3) mAb (BioLegend), and PE/Cyanine7 mouse anti-human CX3CR1 mAb (BioLegend) in another tube. Compensations were established in each staining set using the fluorochromes contained in each staining set. The unstained sample was used as a negative control, and dead cells were detected using 7AAD (BioLegend). The stained cells were measured by a BD FACSCanto II flow cytometer (BD Bioscience, San Jose, CA, USA), and data were analyzed using FlowJo software 10.8.1. (FlowJo, Ashland, OR, USA).

### 2.4. Gating Methods

At first, we used forward scatter and side scatter to gate the population of lymphocytes, followed by classification of single cells, and then 7AAD-negative CD3-expressing cells were categorized into two groups using CD4 and CD8 (Appendix A). For analysis of memory CD4^+^ or CD8^+^ T cells, we assessed CD45RA^−^CD27^+^CD127^+^ T cells as T_CM_ and CD45RA^−^CD27^−^CD127^+^ T cells as T_EM_ in the present study (Appendix A) [16,18]. For analysis of exhausted and differentiated T cells, the expression levels of PD-1, TIM-3, and CX3C chemokine receptor 1 (CX3CR1) were evaluated in CD4^+^ and CD8^+^ T cells (Appendix A). In the present study, we assessed CX3CR1^+^ T cells as differentiated T cells (T_diff_) [19,20].

### 2.5. Enzyme-Linked Immunosorbent Assay (ELISA)

Plasma magnesium (Mg^2+^) concentrations were evaluated using Magnesium Assay Kit (abcam, Cambridge, UK).

### 2.6. Statistical Analysis

The statistical analyses were performed using Graph Pad Prism 9 (Graph Pad Software, San Diego, CA, USA). Two groups were compared using the paired or unpaired Student *t*-test, and multiple groups were compared by one-way analysis of variance followed by a Tukey’s post hoc test. All error bars indicate mean ± standard deviation, and a value of *p* < 0.05 was considered to be significant.

## 3. Results

### 3.1. Higher Frequency of Exhausted T Cells in cStage IV Patients with ESCC

Fifty-five ESCC patients were enrolled in the present study, and their clinical characteristics are shown in Table 1. Neoadjuvant therapy for each patient with cStage II or III and first line treatment for each patient with cStage IV are shown in Table 2. We assessed CD45RA^−^CD27^+^CD127^+^ T cells as T_CM_, CD45RA^−^CD27^−^CD127^+^ T cells as T_EM_, CX3CR1^+^ T cells as T_diff_, and the expression of PD-1 and TIM-3 as markers for exhausted T cells in the present study (Appendix A). No apparent trend in the frequencies of T_CM_, T_EM_, or T_diff_ in either CD4^+^ or CD8^+^ T cells was observed among cStages (Figure 1a). On the other hand, the frequencies of PD-1^+^ or TIM-3^+^CD4^+^ T cells were significantly higher in the cStage IV patients than in any other patients (Figure 1b).

### 3.2. Lower T_CM_ and Higher Exhausted T-Cell Frequencies after CRT

Next, we examined CT/CRT-related alterations in the frequency of each T-cell subset in all of the enrolled patients. Since cStage I patients in the present study did not receive neoadjuvant therapy before blood collection, those patients were classified as the non-treatment group in this analysis. As a result, the frequencies of CD4^+^ and CD8^+^ T_CM_ in the CRT group were significantly lower in comparison to those in the CT group (Figure 2a). In addition, the frequencies of PD-1^+^CD4^+^ and TIM-3^+^CD8^+^ T cells were significantly higher in the CRT group compared to the other groups (Figure 2b). Taken together, it is likely that irradiation might affect the frequencies of exhausted T cells and T_CM_ subsets.

cStage II or III patients underwent subtotal esophagectomy after neoadjuvant CT or CRT, and histological therapeutic effect was determined using surgical specimens according to the Japanese Classification of Esophageal Cancer, 11th Edition. Although we also evaluated the frequencies of T_CM_, T_EM_, T_diff_, and exhausted T cells after neoadjuvant CT or CRT in patients with cStage II or III, there was no obvious trend between the frequency of each T cell subset and pathological response (Figure 3a,b).

It was recently reported that Mg^2+^ sufficiency supports improved T-cell activity against cancer [21]. Although we evaluated plasma Mg^2+^ concentrations among cStages, treatment strategies, and pathological response to neoadjuvant therapy, most patients were not deficient in plasma Mg^2+^ concentration and no significant differences were observed (Appendix A). It is speculated that Mg^2+^ regulates the effector function of CD8^+^ T cells but is not involved in the frequency of each type of T cells.

NLR has been used as an indicator of chronic inflammation and general immune response, and may contribute to evaluation of tumor response in patients treated with immunotherapy [22]. Therefore, we also evaluated the NLR among cStages, treatment strategies, and pathological response of neoadjuvant therapy. NLR was increased significantly in patients with cStage IV compared to those with cStage I and was also increased significantly in patients treated with CT or CRT compared those with non-treatment (Appendix A).

### 3.3. Frequency of T_CM_ in Patients with Progressive Disease Decreased during the Nivolumab Treatment

In the present study, all cStage IV patients were treated with nivolumab, and tumor responses were evaluated during the nivolumab treatment according to the Response Evaluation Criteria in Solid Tumors guidelines, version 1.1. We collected blood samples before and after nivolumab treatment from 11 cStage IV patients and divided them into the responder group (partial response and stable disease) and the progressive disease (PD) group. The frequencies of both CD4^+^ and CD8^+^ T_CM_ were significantly decreased during the course of nivolumab treatment in the PD group (Figure 4a), while there was no alteration in the responder group. PD-1 expression levels in both the CD4^+^ and CD8^+^ T cells were almost unmeasurable after nivolumab treatment (Figure 4b), which was in line with the findings of a previous report [23].

## 4. Discussion

In the present study, we showed that the frequencies of PD-1^+^ or TIM-3^+^CD4^+^ T cells were significantly higher in patients with cStage IV ESCC compared to those at other cStages (Figure 1b), and the PD-1^+^CD4^+^ and TIM-3^+^CD8^+^ T-cell populations were significantly higher in ESCC patients treated with CRT compared to those in patients who underwent other treatments (Figure 2b). In addition, the frequencies of both CD4^+^ and CD8^+^ T_CM_ were significantly decreased in the PD group during the nivolumab treatment (Figure 4a). These results suggest that the frequency of T-cell subpopulations is affected by tumor progression and treatment.

In chronic inflammatory conditions such as cancer, T cells are exhausted and express exhaustion markers including PD-1 and TIM-3 [24,25]. While pre-exhausted T cells express intermediate levels of PD-1 and cytotoxic T-lymphocyte-associated protein-4 (CTLA-4), terminally exhausted T cells express high levels of TIM-3 and other immune inhibitory receptors in addition to PD-1 and CTLA-4 [24,25]. In the present study, we indicated that the frequencies of PD-1^+^ or TIM-3^+^CD4^+^ T cells were significantly higher in cStage IV ESCC patients (Figure 1b). On the other hand, although the frequencies of PD-1^+^ or TIM-3^+^CD8^+^ T cells tended to increase as the stage advanced, the difference was not significant (Figure 1b). The limited number of patients may be a reason, and we are planning to analyze with more patients. In patients with distant metastasis (cStage IV), T cells may become terminally exhausted T cells due to a prolonged chronic inflammatory condition with cancer-bearing status or heavily treated condition by prior chemotherapy. Actually, patients with cStage IV received more intensive treatment (Table 2) and tended to have higher NLR in the present study (Appendix A). It is assumed that terminally exhausted T cells do not respond sufficiently to PD-1 therapy since they produce lower levels of effector cytokines, such as interferon-γ and tumor necrosis factor-α, and have an attenuated anti-tumor effect [24]. Therefore, it is speculated that ESCC patients without distant metastasis or with low tumor burden are good candidates for PD-1 therapy.

CT and CRT are standard therapeutic strategies for patients with advanced ESCC. The efficacies of CT and CRT are assumed to be affected by T-cell immunity as well as hypoxia and tumor cell sensitivity to chemotherapy in the tumor microenvironment [7,8,26]. It has been reported that CT and CRT can enhance the anti-tumor immune response through multiple mechanisms, including immunogenic tumor cell death [7,8], and Chen et al. reported that increases in peripheral CD4^+^ and CD8^+^ T cells induced by CRT are associated with superior survival in ESCC patients [27]. In the present study, the frequencies of both CD4^+^ and CD8^+^ T_CM_ were significantly lower in ESCC patients treated with CRT compared to the frequency in patients treated with CT (Figure 2a), and NLR was significantly higher in CRT group than CT group (Appendix A). Therefore, the addition of irradiation may affect the distribution of CD4^+^ and CD8^+^ T_CM_ through inflammatory changes. In addition, although it is not an obvious trend, CD8^+^ T_EM_ frequency was increased in patients with cStage IV (Figure 1a). These patients received more intensive CT or CRT (Table 2) and tended to have higher NLR (Appendix A). It is suggested that the frequency of CD8^+^ T_EM_ was increased through immunogenic tumor cell death in association with a prolonged chronic inflammatory condition caused by intensive treatment [7,8,9,10].

It has been reported that the frequency of peripheral T_CM_ was high after anti-PD-1 therapy in responders with melanoma, and the tumor-infiltrating T_CM_ correlated with a favorable response to anti-PD-1 therapy in patients with Merkel cell carcinoma [28,29]. We also indicated that the frequencies of both CD4^+^ and CD8^+^ T_CM_ were significantly decreased in the PD group during the nivolumab treatment in cStage IV ESCC patients (Figure 4a). In addition, previous articles showed that TCM exhibited superior anti-tumor activity compared with TEM or effector T cells, such as Tdiff, in the preclinical animal model [30,31]. Therefore, it is assumed that the frequency of T_CM_ needs to be maintained or increased for anti-PD-1 therapy to be effective. Taken together, the alterations in the frequency of peripheral T_CM_ during nivolumab treatment may be a biomarker to predict its efficacy in ESCC patients.

CX3CR1 was stably expressed on differentiated CD8^+^ T cells in the effector phase [19,20], and increased frequency of circulating CX3CR1^+^CD8^+^ T cells correlated with response to anti-PD-1 therapy in patients with renal cell carcinoma, melanoma, and non-small cell lung cancer [20,32,33]. Therefore, we used CX3CR1 as a marker for T_diff_ in the present study. However, the frequency of CD8^+^ T_diff_ did not differ significantly among cStages, treatment strategies, pathological response to neoadjuvant therapy, and during nivolumab treatment (Figure 1a, Figure 2a, Figure 3 and Figure 4a). These results suggest that the frequency of circulating CD8^+^ T_diff_ is not related to the efficacy of nivolumab treatment as well as CT and CRT in patients with ESCC. On the other hand, the frequency of CD4^+^ T_diff_ was significantly increased in patients with cStage IV (Figure 1a) and during the course of nivolumab treatment in PD group (Figure 4a). Further investigation is necessary to elucidate the role of CX3CR1^+^CD4^+^ T cells in tumor immune response.

## 5. Conclusions

Although there are many markers for T_CM_ [14,15,16,17,18], our results indicate that the alteration of frequency of CD45RA^−^CD27^+^CD127^+^ T_CM_ during nivolumab treatment may be a biomarker to predict its therapeutic efficacy in patients with ESCC.

## Figures and Tables

**Figure 1 cancers-15-03641-f001:**
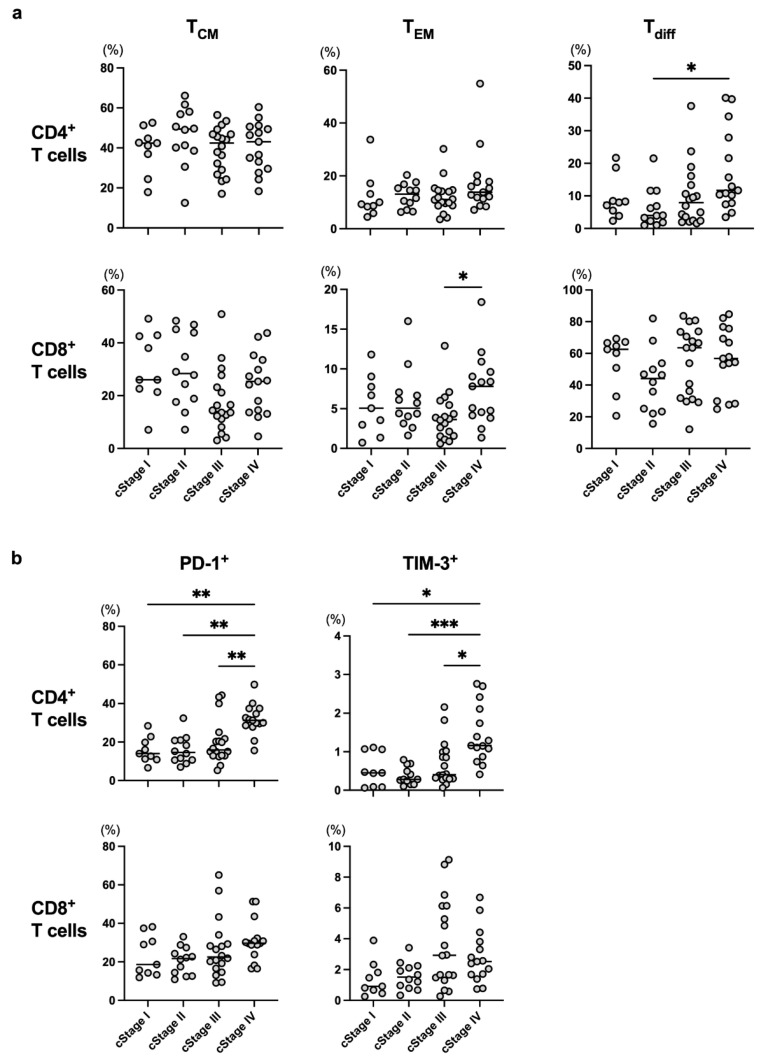
The frequency of each T-cell subset was compared at different cStages. Frequencies of memory T cells (T_CM_ and T_EM_), differentiated T cells (T_diff_) (**a**), and exhausted T cells (**b**) were evaluated among cStages in all enrolled patients. * *p* < 0.05, ** *p* < 0.01, *** *p* < 0.001.

**Figure 2 cancers-15-03641-f002:**
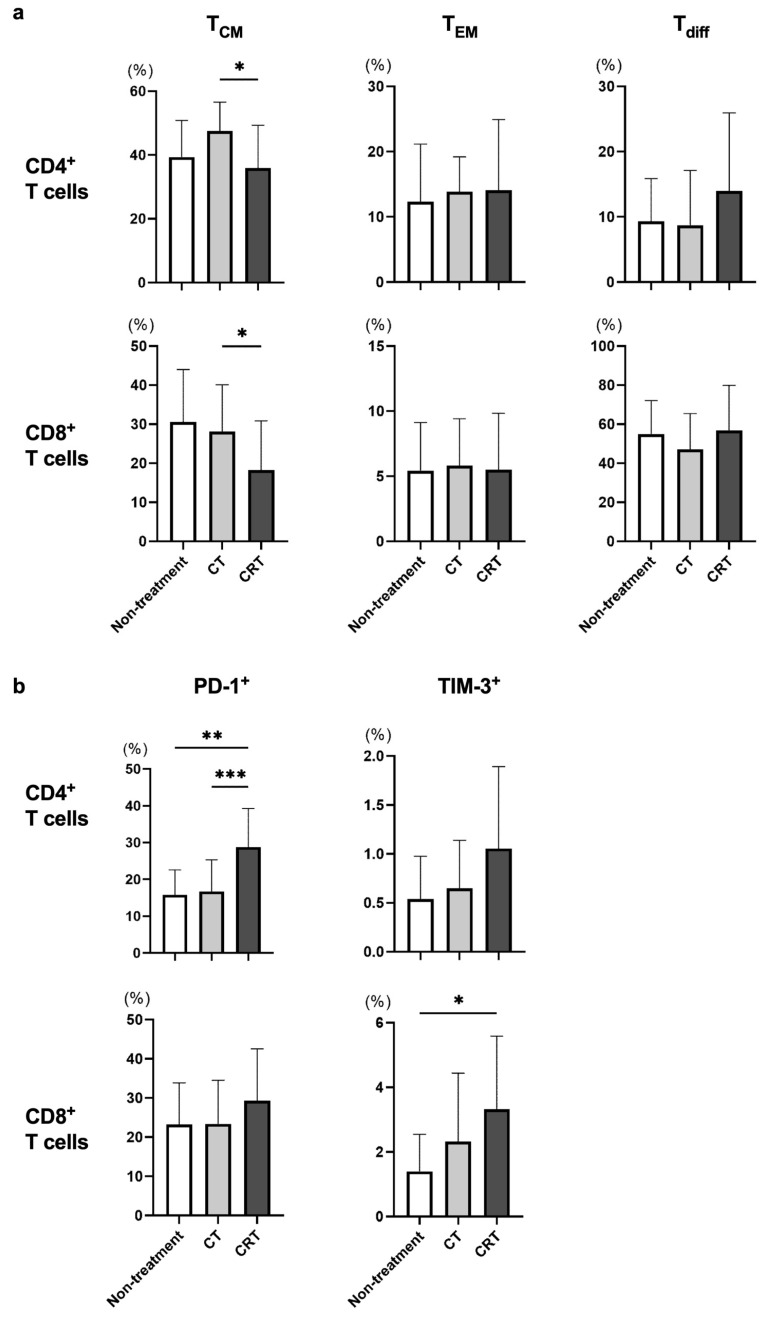
The frequency of each T-cell subset was compared for different treatment strategies. cStage I patients did not receive neoadjuvant therapy before blood collection and were therefore classified as the non-treatment group in this analysis. Frequencies of memory T cells (T_CM_ and T_EM_), differentiated T cells (T_diff_) (**a**), and exhausted T cells (**b**) were evaluated among treatment strategies in all enrolled patients. * *p* < 0.05, ** *p* < 0.01, *** *p* < 0.001.

**Figure 3 cancers-15-03641-f003:**
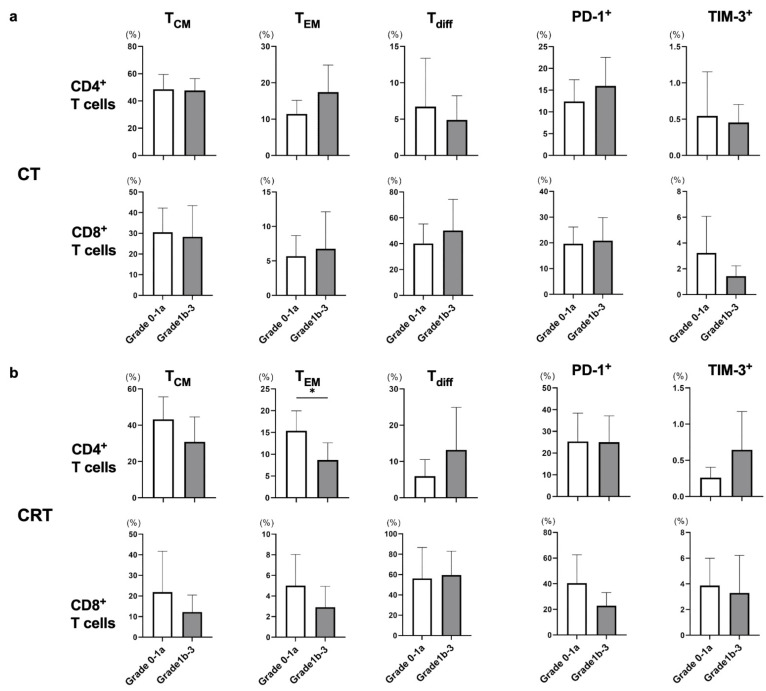
The frequency of each T-cell subset was compared in response to neoadjuvant CT or CRT. The response to neoadjuvant therapy was classified into grade 0–1a and 1b–3 groups, and all cStage II and III patients treated with neoadjuvant CT or CRT were enrolled in this analysis. Frequencies of memory T cells (T_CM_ and T_EM_), differentiated T cells (T_diff_), and exhausted T cells were evaluated between the grade 0–1a and 1b–3 groups in each CT (**a**) and the CRT (**b**) group in eligible patients. * *p* < 0.05.

**Figure 4 cancers-15-03641-f004:**
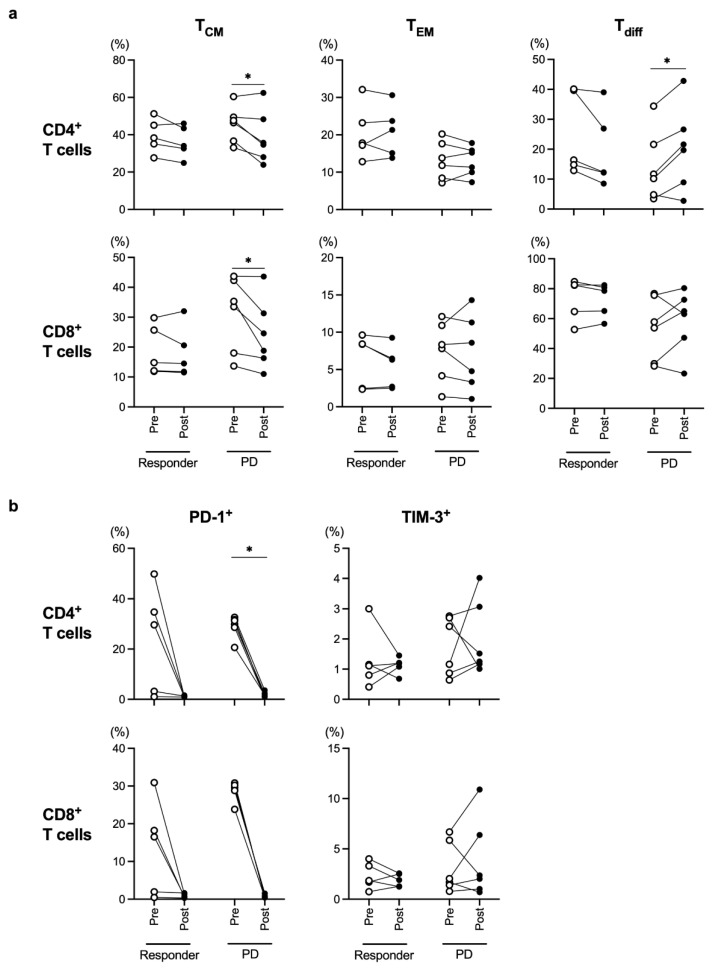
Comparison of each T-cell subset before (Pre) and after (Post) nivolumab therapy in the responder (partial response + stable disease) group and progressive disease (PD) group. Frequencies of memory T cells (T_CM_ and T_EM_), differentiated T cells (T_diff_) (**a**), and exhausted T cells (**b**) were evaluated in all cStage IV patients who were treated with nivolumab therapy. * *p* < 0.05.

**Table 1 cancers-15-03641-t001:** Clinical characteristics of 55 patients with ESCC.

Characteristics		N (% or Median Range)	
Sex			
	Male	42	(76.4)
	Female	13	(22.6)
Age (years)		70	(43–81)
TNM stage *			
	I	9	(16.4)
	II	12	(21.8)
	III	18	(32.7)
	IV	16	(29.1)
Tumor location			
	Upper	7	(12.7)
	Middle	32	(58.2)
	Lower	16	(29.1)
Previous treatment			
	No treatment	9	(16.4)
	Chemotherapy	23	(41.8)
	Chemoradiotherapy	23	(41.8)

* TNM stages were determined according to the Japanese Classification of Esophageal Cancer, 11th Edition.

**Table 2 cancers-15-03641-t002:** Treatments before blood collection and pathological response to neoadjuvant therapy.

cStage	Neoadjuvant Therapy CT (Course) + RT (Gy)	Pathological Response to Neoadjuvant Therapy	cStage	First Line Treatment CT (Course) + RT (Gy)
II	CF (1)	0	IV	S-1 * (3)
	CF (2) + RT (40)	1a		CF (12)
	CF (2)	1b		CF (5) + RT (60)
	CF (1)	1a		CF (1) + RT (40)
	CF (1)	0		CF (1) + RT (40)
	CF (2)	0		CF (5) + RT (60)
	CF (1)	1a		CF (4) + RT (57)
	CF (2)	1a		CF (6)
	CF (2)	2		CF (2) + RT (60)
	CF (1) + RT (40)	3		CF (1) + RT (42)
	CF (2)	3		CF (1) + RT (60)
	CF (2)	2		CF (1) + RT (39)
III	CF (1) + RT (40)	1a		CF (8)
	CF (1) + RT (40)	2		S-1 ** (8)
	CF (1) + RT (40)	1b		CF (2) + RT (60)
	CF (1) + RT (40)	1a		CF (7) + RT (30)
	CF (2)	1b		
	CF (1) + RT (40)	2	RT; radiation therapy	
	CF (1) + RT (40)	1b	CF: 5-fluorouracil 700~800 mg/m^2^ on days 1 to 5, cisplatin 70~80 mg/m^2^ on day 1, every 3–4 weeks	
	CF (2)	1a		
	CF (1) + RT (40)	1b		
	CF (1)	unevaluable	DCF: 5-fluorouracil 750 mg/m^2^ on days 1 to 5, cisplatin 70 mg/m^2^ on day 1, docetaxel 70 mg/m^2^ on day 1, every 3 weeks	
	CF (1) + RT (40)	2		
	CF (1)	1a		
	CF (1) + RT (40)	1b		
	CF (2)	unevaluable	S-1 *: 100 mg/day, orally on days 1 to 14, every 3 weeks	
	CF (2)	1a		
	CF (2) + RT (40)	2	S-1 **: 100 mg/day, orally on days 1 to 28, every 6 weeks	
	DCF (2)	2		
	DadsCF (3)	1a		

* and ** show the difference in S-1 therapy regimens in two different patients. Pathological response to neoadjuvant therapy was determined using surgical specimens according to the Japanese Classification of Esophageal Cancer, 11th Edition.

## Data Availability

The data that support the findings of the present study are available from the corresponding authors upon reasonable request.

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
