# Peer review of "A Potential Biomarker of Dynamic Change in Peripheral CD45RA−CD27+CD127+ Central Memory T Cells for Anti-PD-1 Therapy in Patients with Esophageal Squamous Cell Carcinoma"

_cancers, 2023, doi:10.3390/cancers15143641_

Round 1
Reviewer 1 Report
1- The title is not informative and should be re-written. Authors should present the main part of their result in one sentence or phrase.
2- Abstract's conclusion is not suitable, it is the result iteration and should be re-written.
3- Development of the predicting biomarker was mentioned as the aim of study while assessment of the frequency of memory and exhausted T cells are not known as usual biomarkers, because of complexity of evaluation.
4- Have any consent form was taken from the patients? Please mention
5- Has your study approved by ethical committee? please mention
6- In flow cytometry methods, authors used several fluorochrome conjugate mono-clonal antibodies (almost 10). They should explain how PBMCs were stained in the the related tubes and which isotype controls were used for each dye. The authors should also explain how they compensate and correct the spillovers simultaneously.
7-There are several ambiguities and defects in discussion as below:
a- How authors describe significancy only in CD4+ diff and CD8+ EM T cells (fig 1a)?
b- How authors describe no significancy in CD8+ T cells expressing PD1 and TIM3 in different stages of ESCC (fig 1b)
c- As mentioned in line 164, Mg2+ support improved T cell activity while it seems not any association with the any type of T cells. Regarding, how authors describe no significancy in its plasma concentration between different stages and grades of ESCC patients before and after CT and CRT treatment (fig S3).
d- How authors describe only the frequency of CM T cells (both CD4+ and CD8+) but not EM or Diff T cells were decreased and only the frequency of CD4+ Diff T cells were increased after Nivolumab treatment (Fig 4a)? So, how it could be as a biomarker for Nivolumab treatment efficacy (line 233-235)? It seems it is not a logic deduction! Authors should discuss their own results with other studies.
8- The conclusion is not informative and only contains some results. It should be re-written.
There are some grammatical errors within the manuscript that should be addressed and reconsidered.
Author Response
We would like to appreciate the reviewers for their insightful comments and suggestions. All new additions are highlighted with yellow in the revised manuscript.
Reviewer 1
1- The title is not informative and should be re-written. Authors should present the main part of their result in one sentence or phrase.
Response 1)
We revised the title according to the comment.
2- Abstract's conclusion is not suitable, it is the result iteration and should be re-written.
Response 2)
We revised the conclusion of Abstract.
3- Development of the predicting biomarker was mentioned as the aim of study while assessment of the frequency of memory and exhausted T cells are not known as usual biomarkers, because of complexity of evaluation.
Response 3)
Thank you very much for comment. It has been recently reported that the memory T cells seem to play an important role in anti-tumor immunity (Ref 15, 16 in the text) and the increased frequency of CD8 T cells expressing PD-1 prior to treatment correlated with clinical benefit of PD-1 blockade (Kwon M, et al. Cancer Discov. 2021;11(9):2168-85.). In addition, it is reported that chemotherapy and chemoradiotherapy under certain condition can induce immunogenic tumor cell death, resulting in an activation of cancer immunity cycle including memory and exhausted T cells (Ref 7-10 in the text). As in these reports, T-cell immunity has been reported to be associated with the efficacy of chemotherapy and chemoradiotherapy as well as immunotherapy. We can easily evaluate the frequency of memory and exhausted T cells in peripheral blood collected as a liquid biopsy. Therefore, we have challenged this study in the hope that we could use them as biomarkers to predict each treatment efficacy.
4- Have any consent form was taken from the patients? Please mention
Response 4)
Written informed consent was obtained from all subjects involved in the study. We revised “Informed Consent Statement” in line 319-320.
5- Has your study approved by ethical committee? please mention
Response 5)
We mentioned about it in line 316-318.
6- In flow cytometry methods, authors used several fluorochrome conjugate mono-clonal antibodies (almost 10). They should explain how PBMCs were stained in the the related tubes and which isotype controls were used for each dye. The authors should also explain how they compensate and correct the spillovers simultaneously.
Response 6)
Thank you for the comment. We added the sentence regarding the staining and compensation method of flow cytometry in line 105-119.
7-There are several ambiguities and defects in discussion as below:
a- How authors describe significancy only in CD4+ diff and CD8+ EM T cells (fig 1a)?
Response 7-a)
Thank you for the suggestion. We added the discussion regarding CD8+ EM T cells in line 268-274 and regarding CD4+ diff T cells in line 294-297.
b- How authors describe no significancy in CD8+ T cells expressing PD1 and TIM3 in different stages of ESCC (fig 1b)
Response 7-b)
Thank you for the comment. We added comments regarding CD8+ T cells expressing PD1 and TIM3 in line 245-248.
c- As mentioned in line 164, Mg2+ support improved T cell activity while it seems not any association with the any type of T cells. Regarding, how authors describe no significancy in its plasma concentration between different stages and grades of ESCC patients before and after CT and CRT treatment (fig S3).
Response 7-c)
Thank you for the comment. We added comments regarding Mg2+ in line 188-191.
d- How authors describe only the frequency of CM T cells (both CD4+ and CD8+) but not EM or Diff T cells were decreased and only the frequency of CD4+ Diff T cells were increased after Nivolumab treatment (Fig 4a)? So, how it could be as a biomarker for Nivolumab treatment efficacy (line 233-235)? It seems it is not a logic deduction! Authors should discuss their own results with other studies.
Response 7-d)
We appreciate the crucial suggestion. We revised and added sentences in line 278-283, and added comments regarding CD4+ diff T cells in line 294-297.
8- The conclusion is not informative and only contains some results. It should be re-written.
Response 8)
Thank you for the comment. We revised the conclusion.
Comments on the Quality of English Language
There are some grammatical errors within the manuscript that should be addressed and reconsidered.
Response
We used the English editing service in Fukushima Medical University for revision.
Addition
Based on the comments of other reviewers, we have added results of neutrophil-to-lymphocyte ratio (NLR) analysis and provided a list of treatments administered prior to the blood collection to consider the impact of chemotherapy and chemoradiotherapy on the immune response (Supplementary Figure S4 and Table 2). We have added comments on NLR and the possible impact of treatment intensity on T cells in line 100-103, 192-198, 250-252, 266-274, 306-307, respectively.
Reviewer 2 Report
To develop a set of biomarkers that would predict the efficacy of chemo-, chemoradio- and anti-PD-1 immune checkpoint blockade therapies, the authors have analyzed the subsets of T cells, including Tcm, Tem and Tdiff of both CD4 and CD8 T cells, and frequency of exhaustion markers (PD-1+ and TIM-3+) in these cells in correlation with tumor progression and in the efficacy of anti-PD-1 therapy in patients with esophageal squamous cell carcinoma. They have collected a good set of data to show that the frequency of exhausted T cells is affected by tumor progression and frequency of Tcm is involved in the efficacy of anti-PD-1 therapy.
Strength of the study: In those subsets of T cells examined, they have shown some good correlations. However, it is purely descriptive.
Weakness of the study: In the tumor microenvironment, there are other types of immune cells that may play major roles in tumor progression and therapeutic outcomes. The correlations between these markers in macrophages and other immune cells may be worth looking at.
In summary, this study found some biomarkers in T cell subsets that would predict tumor progression and outcomes of multiple therapeutic modalities.
Minor points:
1. How the patients were treated was not described at all or not adequately described. These patients were treated with CT, CRT, or nivolumab, but without describing dose, frequency, duration, ect. This makes it difficult for others to reproduce the results if needed.
2. The authors do not say if the patients’ written consent for patient sample analysis was conducted.
3. There are minor errors/misses in the following references:
Ref #6. Volume and page numbers?
Ref #22. Article number?
Ref #30. Article number?
Author Response
We would like to appreciate the reviewers for their insightful comments and suggestions. All new additions are highlighted with yellow in the revised manuscript.
Reviewer 2
To develop a set of biomarkers that would predict the efficacy of chemo-, chemoradio- and anti-PD-1 immune checkpoint blockade therapies, the authors have analyzed the subsets of T cells, including Tcm, Tem and Tdiff of both CD4 and CD8 T cells, and frequency of exhaustion markers (PD-1+ and TIM-3+) in these cells in correlation with tumor progression and in the efficacy of anti-PD-1 therapy in patients with esophageal squamous cell carcinoma. They have collected a good set of data to show that the frequency of exhausted T cells is affected by tumor progression and frequency of Tcm is involved in the efficacy of anti-PD-1 therapy.
Strength of the study: In those subsets of T cells examined, they have shown some good correlations. However, it is purely descriptive.
Weakness of the study: In the tumor microenvironment, there are other types of immune cells that may play major roles in tumor progression and therapeutic outcomes. The correlations between these markers in macrophages and other immune cells may be worth looking at.
In summary, this study found some biomarkers in T cell subsets that would predict tumor progression and outcomes of multiple therapeutic modalities.
Response)
Thank you very much for the comment. As the reviewer mentioned, we agree that immune suppressive cells, such as regulatory T cells and M2 tumor associated macrophages, in the tumor microenvironment have a significant impact on tumor growth and therapeutic outcome. However, we have attempted to develop the liquid biopsy predicting the therapeutic efficacy in patients with esophageal squamous cell carcinoma and evaluated the phenotype of T cells in peripheral blood in this study. Since we wish to focus on liquid biopsy in this study, we have added an evaluation of neutrophil-to-lymphocyte ratio (NLR) instead of analysis of the tumor microenvironment in the revised manuscript. Because NLR has been used as an indicator of chronic inflammation and general immune response, and may contribute to evaluation of tumor response in patients treated with immunotherapy. We have added new Supplementary Figure S4 and comments in line 100-103, 192-198, 250-252, 266-274, 306-307.
Minor points:
- How the patients were treated was not described at all or not adequately described. These patients were treated with CT, CRT, or nivolumab, but without describing dose, frequency, duration, ect. This makes it difficult for others to reproduce the results if needed.
Response 1)
We appreciate the crucial suggestion. We have provided a list of treatments administered prior to the blood collection to consider the impact of chemotherapy (CT) and chemoradiotherapy (CRT) on the immune status (new Table 2). We have added comments on the possible impact of treatment intensity on T cells in line 145-146, 250-252, 270-271.
- The authors do not say if the patients’ written consent for patient sample analysis was conducted.
Response 2)
Written informed consent was obtained from all subjects involved in the study. We revised “Informed Consent Statement” in line 319-320.
- There are minor errors/misses in the following references:
Ref #6. Volume and page numbers?
Ref #22. Article number?
Ref #30. Article number?
Response 3)
Thank you for the comment. We have checked and revised them.
Reviewer 3 Report
Esophageal carcinoma presents a poor prognosis. It is imperative to elucidate biomarkers for prognostication and investigate the mechanisms of chemotherapy. The rational course of action entails developing innovative biomarkers in the peripheral blood. However, certain aspects warrant meticulous consideration. Discrepancies were observed in the points of blood examination. In patients with cStage II and III, neoadjuvant treatment with CT or CRT was administered, followed by the collection of blood samples prior to surgery. In patients with cStage IV, peripheral blood samples were obtained after the initiation of first-line therapy with CT or CRT, and before the administration of nivolumab as second-line treatment. Anticipatedly, a high frequency of exhausted T cells was expected in cStage IV patients due to the impact of chemotherapy, representing a distinct condition in comparison to cStage II and III patients. The sample size in each group was limited, and the study design may have been premature. The authors ought to contemplate revising the manuscript in light of the ensuing commentary.
Major Comment:
1. No data is provided regarding the relationship between peripheral blood and immune status in the tumor. As suggested by the authors, immunohistochemistry analysis of surgical specimens may offer insights into divergent immune statuses. Additional multifaceted supportive analyses are imperative.
2. Authors described that the frequency of both CD4+ and CD8+ TCM was significantly decreased during the course of nivolumab treatment in PD group (Fig. 4a). Moreover, PD-1 expression on both CD4+ and CD8+ T cells was almost unmeasurable after nivolumab treatment (Fig. 4b). Although the statistical results appear supportive, the data appears somewhat scattered. This result could be influenced by the different time points of blood sampling. It is challenging to compare the data from different time points.
Author Response
We would like to appreciate the reviewers for their insightful comments and suggestions. All new additions are highlighted with yellow in the revised manuscript.
Reviewer 3
Esophageal carcinoma presents a poor prognosis. It is imperative to elucidate biomarkers for prognostication and investigate the mechanisms of chemotherapy. The rational course of action entails developing innovative biomarkers in the peripheral blood. However, certain aspects warrant meticulous consideration. Discrepancies were observed in the points of blood examination. In patients with cStage II and III, neoadjuvant treatment with CT or CRT was administered, followed by the collection of blood samples prior to surgery. In patients with cStage IV, peripheral blood samples were obtained after the initiation of first-line therapy with CT or CRT, and before the administration of nivolumab as second-line treatment. Anticipatedly, a high frequency of exhausted T cells was expected in cStage IV patients due to the impact of chemotherapy, representing a distinct condition in comparison to cStage II and III patients. The sample size in each group was limited, and the study design may have been premature. The authors ought to contemplate revising the manuscript in light of the ensuing commentary.
Response)
We appreciate the crucial comment. We agree with a reviewers’ opinion that intense chemotherapy (CT) and chemoradiotherapy (CRT) may affect the immune status. We have provided a list of treatments administered prior to the blood collection to consider the impact of CT and CRT on the immune status (Table 2). We have added a comment on the possible impact of treatment intensity on T cells in line 145-146, 250-252, 270-271.
Major Comment:
- No data is provided regarding the relationship between peripheral blood and immune status in the tumor. As suggested by the authors, immunohistochemistry analysis of surgical specimens may offer insights into divergent immune statuses. Additional multifaceted supportive analyses are imperative.
Response 1)
Thank you very much for the comment. As the reviewer mentioned, we agree that immune status in the tumor is also important for immunotherapy. However, we have attempted to develop the liquid biopsy predicting the therapeutic efficacy in patients with esophageal squamous cell carcinoma and evaluated the phenotype of T cells in peripheral blood in this study. Since we wish to focus on liquid biopsy in this study, we have added an evaluation of neutrophil-to-lymphocyte ratio (NLR) instead of analysis of the tumor microenvironment in the revised manuscript. Because NLR has been used as an indicator of chronic inflammation and general immune response, and may contribute to evaluation of tumor response in patients treated with immunotherapy. We have added Supplementary Figure S4 and comments in line 100-103, 192-198, 250-252, 266-274, 306-307.
- Authors described that the frequency of both CD4+ and CD8+ TCM was significantly decreased during the course of nivolumab treatment in PD group (Fig. 4a). Moreover, PD-1 expression on both CD4+ and CD8+ T cells was almost unmeasurable after nivolumab treatment (Fig. 4b). Although the statistical results appear supportive, the data appears somewhat scattered. This result could be influenced by the different time points of blood sampling. It is challenging to compare the data from different time points.
Response 2)
Thank you for the comment. Recently, it has been reported that dynamic change of each T cell phenotype, which compares samples collected at different time points during treatment, is useful for predicting the efficacy of CRT and immune checkpoint inhibitors (Ref 20, 26 in the text. Wei H, et al. Frontiers in Immunology. 2022;13:1060695.). Therefore, we have tried to analyze the dynamic change in memory and exhausted T cells during nivolumab treatment in this study. As noted in the reviewer's comments, differences in the time points of blood sampling may affect the frequency of memory and exhausted T cells. We added the comment regarding timing of blood collection during nivolumab treatment in line 95-97.
Round 2
Reviewer 1 Report
Dear authors
Your revised version is now acceptable.
It is now acceptable
Reviewer 3 Report
Thank you for revising the article in accordance with the suggestion. No further requisites are necessary.